# Adding evidence of the effects of treatments into relevant Wikipedia pages: a randomised trial

Clive E Adams ,[1] Alan A Montgomery,[2] Tony Aburrow,[3] Sophie Bloomfield,[4] Paul M Briley ,[1] Ebun Carew,[5] Suravi Chatterjee-Woolman,[6] Ghalia Feddah,[7] Johannes Friedel,[8] Josh Gibbard,[9] Euan Haynes,[10] Mohsin Hussein,[11] Mahesh Jayaram ,[12] Samuel Naylor,[7] Luke Perry,[13] Lena Schmidt,[14,15] Umer Siddique,[16] Ayla Serena Tabaksert,[17] Douglas Taylor,[18] Aarti Velani,[19] Douglas White,[20] Jun Xia[21,22]

For numbered affiliations see end of article.

**Correspondence to**
Clive E Adams;
clive.adams@nottingham.ac.uk

## ABSTRACT

**Objectives** To investigate the effects of adding high-grade quantitative evidence of outcomes of treatments into relevant Wikipedia pages on further information-seeking behaviour by the use of routinely collected data.

**Setting** Wikipedia, Cochrane summary pages and the Cochrane Library.

**Design** Randomised trial.

**Participants** Wikipedia pages which were highly relevant to up-to-date Cochrane Schizophrenia systematic reviews that contained a Summary of Findings table.

**Interventions** Eligible Wikipedia pages in the intervention group were seeded with tables of best evidence of the effects of care and hyperlinks to the source Cochrane review. Eligible Wikipedia pages in the control group were left unchanged.

**Main outcome measures** Routinely collected data on access to the full text and summary web page (after 12 months).

**Results** We randomised 70 Wikipedia pages (100% follow-up). Six of the 35 Wikipedia pages in the intervention group had the tabular format deleted during the study but all pages continued to report the same data within the text. There was no evidence of effect on either of the coprimary outcomes: full-text access adjusted ratio of geometric means 1.30, 95% CI: 0.71 to 2.38; page views 1.14, 95% CI: 0.6 to 2.13. Results were similar for all other outcomes, with exception of Altmetric score for which there was some evidence of clear effect (1.36, 95% CI: 1.05 to 1.78).

**Conclusions** The pursuit of fair balance within Wikipedia healthcare pages is impressive and its reach unsurpassed. For every person who sought and clicked the reference on the 'intervention' Wikipedia page to seek more information (the primary outcome), many more are likely to have been informed by the page alone. Enriching Wikipedia content is, potentially, a powerful way to improve health literacy and it is possible to test the effects of seeding pages with evidence. This trial should be replicated, expanded and developed.

**Trial registration number** IRCT2017070330407N2.

## BACKGROUND

Wikipedia is a free-content online encyclopaedia containing articles on a vast range of topics.[1] At present, there are over 5.7 million articles, 46 million pages in the English language.[2] Since its creation in 2001, Wikipedia has expanded to attract over 27 million registered users[3] with 16 billion page views per month.[4] This made Wikipedia the fifth most popular site on the internet in 2017.[5]

Wikipedia is openly editable. This means that any one of these users can access *and edit* the majority of articles. Wikipedia policy states, however, that all information presented in pages must be 'verifiable against a published reliable source'.[1] Therefore, all pages aim to contain references for the information they provide. To prevent the risk of pages being devalued with misinformation, Wikipedia has various quality control measures. These include a 'watchlist' to notify editors when a page has been edited, a published list of recent changes that editors can access to review, automated computer scripts, page protection on more controversial pages, edit filters on certain pages and blocking any editors who repeatedly damage the value of the page.[6] On top of this, Wikipedia has a team of administrators. They are editors who have been given access to additional tools on

their account. These include the ability to block/unblock accounts, edit fully protected pages and delete/undelete pages. There are 1194 administrators on the English language Wikipedia (as of December 2018).[2]

## Strengths and limitations of this study

► First randomised trial of placement of evidence within Wikipedia pages.
► The use of routine data to allow 100% follow-up.
► Open editing of Wikipedia pages—both intervention and control pages—by the Wikipedia community served to minimise difference between groups.
► Outcomes necessitated unusual levels of interest and commitment on the part of the Wikipedia page reader.
► Small study in highly specialised area of heath care.

Wikipedia contains many pages relating to healthcare. In 2014, the English language version was estimated to contain 25 000 articles on health-related topics, while across all languages, there are 155 000 articles containing 950 000 references.[7] These are often accessed via search engine results with one survey suggesting that around 22% of healthcare-related online searches direct to Wikipedia pages.[8 9] In 2013, health pages on Wikipedia received 4.8 billion views, making it one of the most used means for accessing health information globally.[10] When the use of Wikipedia is studied in medical students and doctors, it is clear that it is becoming an increasingly popular resource.[11 12] This is, perhaps, enhanced by Wikipedia being entirely free of charge—including data download charges in low-income and middle-income countries. In this context, there is criticism that as Wikipedia is openly editable, the information it contains may be unreliable. Some evidence suggests, however, that there is no difference in accuracy when Wikipedia is compared with other professionally maintained medical databases[13] although opinions differ by subspecialty, depend on the 'target' readership and vary across time (table 1).

The Cochrane Collaboration[14] is a non-profit nongovernmental organisation producing, and maintaining systematic reviews of healthcare published within the *Cochrane Library* (by John Wiley). The Collaboration is made up of subgroups and Cochrane Schizophrenia produces and updates high-quality systematic reviews and meta-analyses relevant to people with schizophrenia and related psychotic conditions.[15] In 2004, a group called WikiProject Medicine was started with the aim of creating and managing medical articles on Wikipedia. This group allows discussion and collaboration on these articles to improve the quality of the information presented.[6] In

**Table 1** Selection of studies of Wikipedia's value to different readerships by medical subspecialty

| Subspecialty (reference) | Date | Assessing for suitability for … | Conclusion |
|---|---|---|---|
| Ten most costly conditions[38] | 2014 | General readership | Most Wikipedia articles representing the 10 most costly medical conditions (…) contain many errors when checked against standard peer-reviewed sources. Caution should be used (…) |
| Cancer—general[13] | 2011 | Patients | Wiki resource had similar accuracy and depth as the professionally edited database |
| Cancer—osteoscarcoma[39] | 2010 | Patients | (…) the quality of osteosarcoma-related information found in the English Wikipedia is good but inferior to the patient information provided by the National Cancer Institute |
| Cardiovascular[40] | 2015 | Medical students | Wikipedia entries are not aimed at a medical audience and should not be used as a substitute to recommended medical resources. Course designers and students should be aware that Wikipedia entries on cardiovascular diseases lack accuracy, predominantly due to errors of omission. |
| Complementary medicine[41] | 2014 | General readership | Patients and health professionals should not rely solely on Wikipedia for information on these herbal supplements when treatment decisions are being made. |
| Gastro—hepatology[42] | 2014 | Medical students | … not good source of evidence |
| Mental health[43] | 2012 | General readership | The quality of information on depression and schizophrenia on Wikipedia is generally as good as, or better than, that provided by centrally controlled websites, Encyclopaedia Britannica and a psychiatry textbook. |
| Nephrology[44] | 2013 | Patients | Fairly reliable medical resource |
| Orthognathic surgery[45] | 2012 | Patients | Maximum (…) score(ings in comparison to other online sources) were Wikipedia |
| Pharmacology[46] | 2017 | Doctors | Wikipedia lacks the accuracy and completeness of standard clinical references and should not be a routine part of clinical decision making. |
| Pharmacology[47] | 2014 | Medical students | … Wikipedia is an accurate and comprehensive source of drug-related information for undergraduate medical education. |
| Pharmacology[48] | 2008 | Patients | Wikipedia has a more narrow scope, is less complete and has more errors of omission than the comparator database. Wikipedia may be a useful point of engagement for consumers, but is not authoritative and should only be a supplemental source of drug information. |
| Respiratory medicine[49] | 2015 | Medical students | Most articles had knowledge deficiencies, were not accurate and were not suitable for medical students as learning resources. |

2014, a formalised partnership between Wikipedia and Cochrane was created, aiming to 'transform the quality and content of health evidence available online'.[16] This involves incorporating Cochrane's evidence into Wikipedia articles and improving the information's accuracy and reliability.

While increasing accessibility of highest grade maintained healthcare information seems a laudable aim, objective quantification of the effects of this effort has not been undertaken. This paper reports a collaboratively designed pragmatic randomised trial of adding evidence of the effects of care to Wikipedia health pages on the routinely collected indicators of readers' interest.

## AIMS
To evaluate the effects of enriching Wikipedia content with summary tables from level 1 evidence on the effects of care.

## METHODS
In preliminary work, we tested stability of target pages in Wikipedia. Adding an evidence-table to four Wikipedia pages (trifluoperazine—a less used antipsychotic, eg, 3529±198 prescriptions/month—figures are for 2018, NHS England[17]; chlorpromazine—a old widely used antipsychotic drug: 22 386±803 prescriptions/month; palperidone—an expensive new antipsychotic drug: 853±34 prescriptions/month and one important talking therapy—cognitive behavioural therapy). These all four pages remained stable over a 12-month period (2015). Further work investigated what proportion of the topics of Cochrane Schizophrenia reviews already had a highly specific page in Wikipedia. In 2016, around half of Cochrane Schizophrenia reviews had an obvious 'landing' page directly addressing the topic of the review.[18] Then in 2016, we held a 1-day meeting of student volunteers (medicine and students of applied health sciences), trialists and representatives from Wikipedia and John Wiley, to plan this trial.[19] The study is a two-arm, parallel, open, randomised controlled trial with a 1:1 allocation ratio.

The aim of this study was to evaluate the impact of seeding relevant Wikipedia pages with evidence from high-grade systematic reviews on information-seeking behaviour.

## Eligibility
### Inclusion criteria—'participants'
A Wikipedia page which was clearly relevant to an up-to-date Cochrane Schizophrenia systematic review and that review contained at least one Summary of Findings (SoF) table. These tables, created within the GradePro[20] system, are succinct summaries of the key outcomes of the review (box 1).

### Box 1 PICO box

**Lists participants, interventions, controls and outcomes (PICO)**
**P:** Wikipedia pages of direct relevance to up-to-date systematic reviews of the Cochrane Schizophrenia Group.
**I:** Posting the relevant Cochrane review's Summary of Findings table (modified to increase readability) on the target Wikipedia page along with references to the review's web page and full text.
**C:** Leaving the existing page unmodified.
O: Activity on Cochrane web (summary) page specific to that review—thorough the use of Google Analytics—and interest in full Cochrane review—through quantification of full-text downloads and Altmetric scores of social media activity—though routine data supplied by John Wiley. All at 12 months.

### Exclusion criteria
If a highly relevant Wikipedia page existed but the Cochrane review was out of date (a judgement made by CEA), these Wikipedia pages were not included in the trial. Also, we did not create a brand new Wikipedia page, should one have not existed for an up-to-date review.[18] Finally, a specialist review such as 'yoga for schizophrenia' would have been be out of place on a general Wikipedia page about 'yoga' and therefore that more general Wikipedia page was also ineligible.

### Randomisation
Reviews were stratified according to type of intervention (drug or other) and amount of access activity in the year prior to baseline (low or high, according to median split). The latter used Google Analytics' 'pageviews' statistic regarding Cochrane's universally accessible individual review pages.[21] The reviews were then allocated to the intervention or control arm by one of the coauthors (AAM) using a computer-generated random number sequence. Allocation was conducted using unique code numbers for each review rather than review title, to avoid risk of selection bias.

### Interventions
#### Experimental group—interventions
Reviews in the intervention group had a referenced table(s) automatically generated by the use of SEED.[22] This open access software, especially created for this study, uses the original Cochrane review file and rewrites the Cochrane SoF tables in plain English and generates hyperlink references (to both full subscription review and the universally accessible web summary page) (figure 1[23]).

In the design process of our tables, we communicated with members of 'Sense about Science'[24] and consulted publications of the Cochrane Effective Practice and Organisation of Care group[25] in order to increase clarity and readability of the evidence in our tables. More details on how we worked to increase readability are described in the protocol,[19] as well as our publication of the SEED tool.[22] SEED deposits this code in the computer's memory in seconds. The intervention group's Wikipedia editor

| Paliperidone palmitate long-acting injection compared to risperidone for schizophrenia[2] | | | |
|---|---|---|---|
| When flexibly dosed every four weeks, paliperidone palmitate appears comparable in efficacy and tolerability to risperidone. In short-term studies, paliperidone palmitate – the longer-acting injection – has a similar adverse effect profile to related compounds such as risperidone by mouth. No difference was found in the high rate of reported adverse sexual outcomes and paliperidone palmitate is associated with an increase in serum prolactin.[2] | | | |
| [hide] **Outcome** | **Findings in words** | **Findings in numbers** | **Quality of evidence** |
| **Global state: No clinically important change** | | | |
| No 30% improvement on PANSS score. Follow-up: 13-53 weeks | There is no clear difference between people given paliperidone palmitate and those receiving risperidone for this outcome. These findings are based on data of low quality. | RR 1.03 (0.93 to 1.14) | Low |
| **Relapse** | | | |
| Recurrence of psychotic symptoms. Follow up: 13-53 weeks | There is no clear difference between people given paliperidone palmitate and those receiving risperidone for the outcome of 'relapse'. Data supporting this finding are based on moderate quality evidence. | RR 1.23 (0.98 to 1.53) | Moderate |
| **Leaving the study early** | | | |
| - For any reason. Follow up: 13-53 weeks | Paliperidone palmitate causes little or no increase to the chance of leaving the study. | RR 1.12 (1 to 1.25) | High |

**Figure 1** Sample of embedded table.

(LS and JF) had only to paste this code into the Wikipedia page in the relevant subsection for the table and hyperlink to appear. This was undertaken across the second week of July 2017.

All content posted in the scope of this trial was sourced from peer-reviewed, systematic reviews published in the Cochrane Library. It complied with WP:MEDRS quality standards for reliable sources in medicine.[26] The content posted was intended to improve the encyclopaedia's content, complying with its terms of use. The WP:NOTLAB policy[27] outlines disruptive editing and controversial research. We made an effort to be non-disruptive through discussions with Wikipedia representatives before editing content, as well as using solely verifiable, accessible and reliable sources. We did not interfere in cases where the restructuring of Wikipedia articles caused the removal, migration or adaptation of our content, and discuss these cases in our results section.

### Control intervention—control
The control group Wikipedia pages did not have a table or reference added—although seven of these pages already had the Cochrane reference employed. This reference was not removed.

### Source of data—outcomes
The routine data on full review access are collected by the Cochrane Library's publisher, Wiley. These data, kindly supplied by the Cochrane Office John Wiley, report full-text downloads, and Altmetric scores. The latter is a composite weighted measure of the influence of published work online and via social media platforms—in this case composed from monitoring 17 different platforms/news outlets[28] (full list of platforms, and data-by-platform available in data file at https://doi.org/10.17605/OSF.IO/K2SP4). The full review is widely accessible[29] but not universally so. Neither is the full review succinct. However, Cochrane Summaries web pages are both universally accessible and succinct and have been awarded for their use of plain English.[30] They were monitored using the standard (free) service from Google Analytics.[21]

### Outcomes
All outcomes were measured at 12 months. There were two outcomes of coprimary interest:
1. The number of visits to the free summary page (all page views).
2. The number of full-text downloads.

We selected these as the design team[19] felt they represented the best, measurable, most generic indicators of 'more interest' in the evidence as presented in the tables. The first was universally achievable as the web page for each review is free online. The second—the number of full-text downloads—is only possible where this level of access is available. Although coverage of this open service is now considerable,[29] this would, nevertheless, mean that some interested readers may not have been registered because of limited access to that outcome. We have no data for this. Secondary outcomes were divided into activity on the free to all summary page, and outcomes relating to activity on the Cochrane Library's full review. More subtle but potentially relevant effects, such as effect on reader behaviour or information comprehension were beyond the scope of the methods used.

## Statistical considerations

The sample size for this study is fixed by the number of eligible Wikipedia pages and Cochrane reviews. From preliminary work we had expected to be able to randomise around 100 pages,[18] enabling detection of a between-group standardised difference of 0.57 with 80% power and 5% two-sided alpha. However, due to some reviews being too out of date to report on Wikipedia, the actual number available was 70 which permits detection of an effect size of 0.68.

We compared characteristics of the intervention and control arms at baseline using descriptive statistics. For all between-group comparisons, we analysed Wikipedia pages as randomised regardless of how long the Wikipedia page held the table. We estimated between-group effects using multivariable linear regression models adjusting for baseline activity, presented with 95% CIs and p-values, and with log-transformation of outcomes as required. For such outcomes, results are presented as ratios of geometric means. Data were analysed using Stata V.15.

## RESULTS

All 70 eligible Wikipedia pages relevant to up-to-date Cochrane reviews were randomised, and complete follow-up data were available for all (figure 2).

At baseline, Altmetric scores were evenly distributed (table 2).

During the study, 14 of the intervention group's references had additional hyperlinked PubMed IDs added, most probably by Wikipedia's automatic updating service bots. Also, six of the 35 intervention group *tables* were removed after 2 months (three pages), 5, 8 and 11 months (one page each) but the *information in the tables* remained within the text as did the hyperlinks (83% of full tables remained 95% CI: 67% to 92%; 100% information remained). As mentioned before, seven of the control pages (20%–95% CI: 10% to 36%) did already have a reference to the relevant Cochrane review. In accordance with WP:NOTLAB policy on minimal disruption to pages,[27] and pragmatic trial design in which even 'control' patients may receive some of the experimental treatment if this is in the course of routine care,[31] this reference was not removed but no table was added.

One review in the control arm had very high page views (25 794, 68x the median for whole sample) but not full-text accesses[32] and one review in the intervention arm had very high full-text accesses (7407, 18x the median for whole sample[33]).

Although the point estimates for the ratio of geometric means favoured the intervention group for both coprimary outcomes, the CIs were wide and there was no statistical evidence of an effect (table 3). Results were similar for secondary outcomes, with the exception of Altmetric score which indicated some evidence of an intervention effect, with 95% CI ranging from 5% to 78% increase in geometric mean.

## DISCUSSION

This is the first randomised trial of Wikipedia content. Randomisation has been employed before to investigate Wikipedia linguistics[34] but not for the effect of placement of evidence within the page. Our design tried to balance needs of end-users, Wikipedia administrators and editors and methodologists. The intervention was the insertion of an evidence table and references (with hyperlinks) to the source systematic reviews into a highly relevant Wikipedia page. This intervention resulted in no clear, statistically significant, difference in access to the full review and page views after 1 year. Although all outcome measures consistently favoured a finding indicating increasing activity on the reviews in the 'intervention' group although only the Altmetric score—a measure of relevant social media activity—reached conventional levels of statistical significance. Inspection of the constituent parts of the composite Altmetric score (please see data file at https://doi.org/10.17605/OSF.IO/K2SP4) gives no indication that the Wikipedia subscore is simply causing the elevation in Altmetric ratings. The elevation seemed more linked to microblogging sites such as Twitter.

Six tables were deleted at different points across the year out of the 35 inserted into Wikipedia pages. Deletion was undertaken after debate with the Wikipedia user and then the Wikipedia Administrator and is part of the evolution of Wikipedia pages. Administrators have to ensure that this is undertaken in a balanced way taking into account the needs of the readership. Although the tables were deleted, the tables' evidence continued to be reported, as were the hyperlinks. To some readers, the tabular format was unacceptable as they felt that tables made the pages 'too academic' in appearance. We felt, however, the table was attractive and informative and might encourage interest as well as the seeking of the hyperlink and using it (our primary outcome). Although, after these edits, the hyperlink remained, we think deletion of the table would probably help approximate the results of experimental and control groups. This also illustrates how Wikipedia

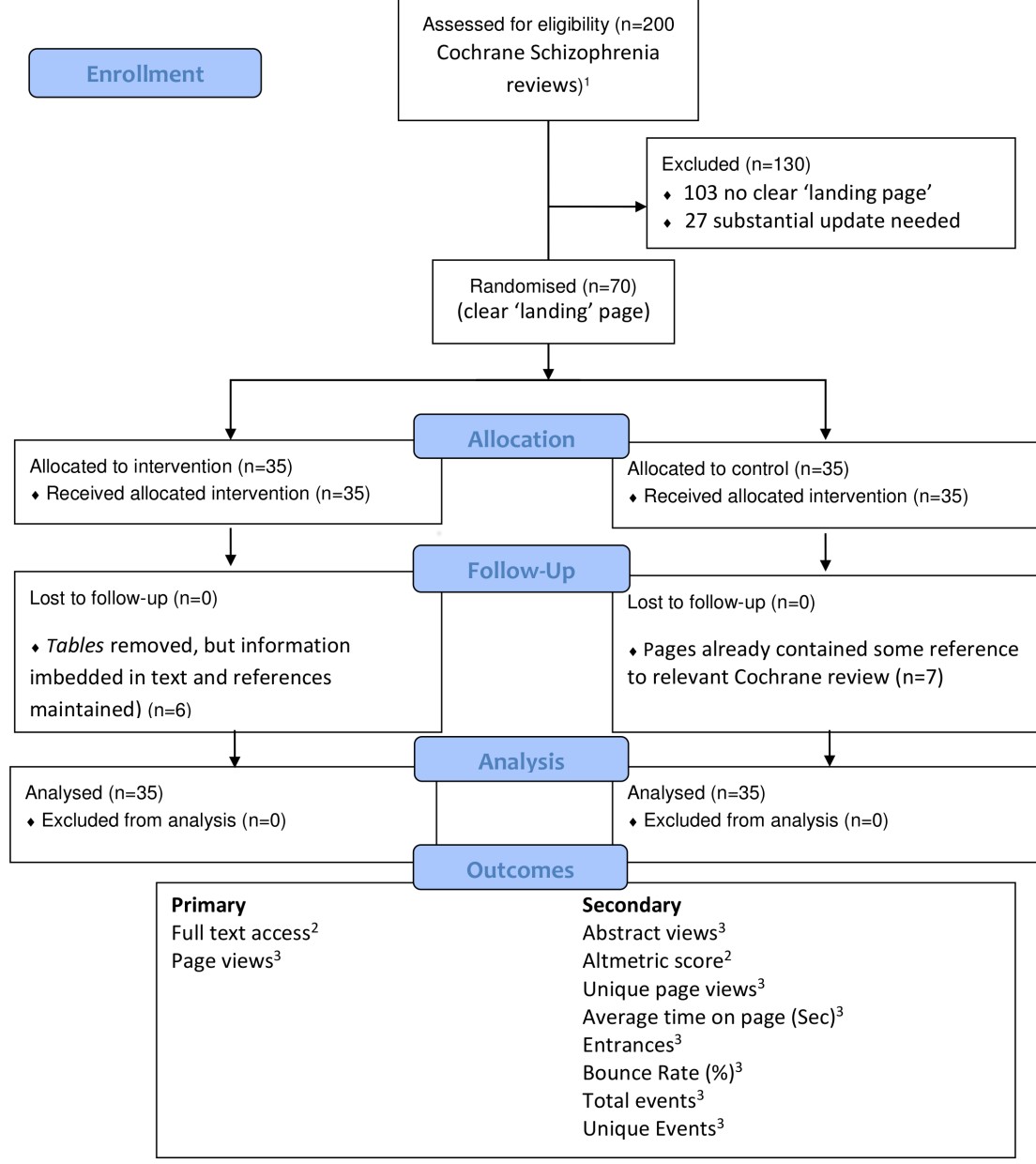

**Figure 2** Consolidated Standards of Reporting Trials flow diagram.

pages evolve across time. End user feedback is considered and balanced compromises are made. The input to any Wikipedia page, even by respected experts, is not sacrosanct and can be edited in ways that some may not consider advantageous to increasing readership. Working with Wikipedia has the attraction of being dynamic but necessitates commitment, and, for those who feel uncomfortable with their work being edited by unknown others, maintaining Wikipedia evidence could be a less rewarding experience.

The addition of the PubMed IDs broadens the options for gaining additional information for users of the Wikipedia page. However for this trial, again, these additions could have served to narrow any difference between

**Table 2** Baseline Altmetric scores

| Group | N | Arithmetic mean | SD | Median | 25th centile | 75th centile | Min | Max |
|---|---|---|---|---|---|---|---|---|
| Control | 35 | 18 | 30 | 10 | 5 | 19 | 2 | 160 |
| Intervention | 35 | 19 | 24 | 12 | 5 | 25 | 2 | 105 |

Max, Maximum; Min, Minimum; N, Number; SD, Standard Deviation.

**Table 3** Results

| Group | N | Arithmetic mean | SD | Geometric mean | Adjusted ratio of geometric means | 95% CI | P value |
|---|---|---|---|---|---|---|---|
| **Coprimary outcomes** | | | | | | | |
| Full-text access | | | | | | | |
| Control | 35 | 654 | 721 | 331 | – | – | |
| Intervention | 35 | 994 | 1448 | 437 | 1.30 | 0.71 to 2.38 | 0.39 |
| Page views | | | | | | | |
| Control | 35 | 1427 | 4379 | 318 | – | – | |
| Intervention | 35 | 618 | 656 | 366 | 1.14 | 0.60 to 2.13 | 0.69 |
| **Secondary outcomes** | | | | | | | |
| Altmetric score | | | | | | | |
| Control | 35 | 19 | 29 | 11 | – | – | |
| Intervention | 35 | 25 | 32 | 15 | 1.36 | 1.05 to 1.78 | 0.02 |
| Abstract views | | | | | | | |
| Control | 35 | 364 | 368 | 228 | – | – | |
| Intervention | 35 | 441 | 464 | 271 | 1.17 | 0.76 to 1.81 | 0.47 |
| Unique page views | | | | | | | |
| Control | 35 | 1307 | 4032 | 290 | – | – | |
| Intervention | 35 | 561 | 596 | 331 | 1.13 | 0.60 to 2.12 | 0.70 |

CI, Confidence Interval ; N, Number; SD, Standard Deviation.

| Group | N | Arithmetic mean | SD | Adjusted difference in means | 95% CI | P value |
|---|---|---|---|---|---|---|
| Time on page (seconds) | | | | | | |
| Control | 35 | 165 | 69 | – | – | |
| Intervention | 35 | 183 | 76 | 18.51 | –16.06 to 53.08 | 0.29 |

CI, Confidence Interval ; N, Number; SD, Standard Deviation.

intervention and control. Finally, at the very start of the trial, seven of the *control* pages already had some reference to the Cochrane review. Because of our commitment to minimal disruption of the existing Wikipedia pages and to pragmatism in randomised trials,[27 31] we did not feel it right to delete these references but their presence may also have narrowed the gap between intervention and control groups.

There is little similar literature to contextualise this work. We previously conducted an RCT of Cochrane Schizophrenia review engagement after sending short messages containing review titles or pertinent questions/results relevant to the review via the social media platforms Twitter and Weibo.[35] In that study, the primary outcome of increasing views of the review summary page was met, as were several secondary outcomes measuring review engagement (although we did not have data on full-text access or Almetric scores). Importantly, the Twitter study measured further review engagement after the relatively few @CochraneSzGroup and Wiebo followers had received a very short fragment (140 characters) of review information. In the current trial, however, we measured

engagement after providing the 7 331 024 page viewers (figures for year 10 July 2017 to 9 July 2018, calculated using Pageview Analysis[36]) to the 70 Wikipedia pages much more evidence (a concise summary-of-findings table). It is possible that the embedded summary-of-findings table may have satiated more readers' appetites for evidence at the time of reading and may have *reduced* the impulse to click out. Also, in the Twitter trial, the 'target' page was one click away. In this Wikipedia trial, the reader had to undertake a minimum of two clicks. Although this difference sounds minimal, it does indicate a considerable commitment of the reader to pursue more information. In this trial, for an outcome to occur, the Wikipedia user had usually to scroll down to find the table, click to expand the drop-down format of the table, seek the reference to that table and finally click out on one of the hyperlinks. This complex set of actions would, we suggest, indicate high levels of motivation to seek further information and it would seem likely that many users of the Wikipedia pages would have not gone further than the initial page. The Twitter trial suggested a large effect on information-seeking behaviour in a small population,

this Wikipedia study did suggest a modest effect—but on a very large population—and in doing this, is important. Many refinements and improvements of this Wikipedia intervention are possible and testable.

Evaluating techniques of dissemination of knowledge is entirely possible and urgent as calls for efficient use of ever-more platforms increase. Much effort may be squandered on attractive but ineffective ideas. This first trial of placement of evidence within Wikipedia supports the need for more evaluative studies of this particular platform. Although only one secondary outcome reached conventional levels of statistical significance, all outcomes did favour—to some extent—the Wikipedia pages seeded with evidence tables (consistent potential 13%–36% increase in activity across all findings). We think this supports the hypothesis that seeding Wikipedia with evidence could be a potent way of encouraging readers to seek more in-depth information on the effects of care. The hit-rate on the 70 very highly specialised Wikipedia pages was over 500K/month. If even half were the activity of robotic automated systems[37] that still leaves considerable activity from interested people. How best to seed good evidence into Wikipedia, how best to communicate with this readership, how to use images and infographics and how to work with Wikipedia to best advantage of all, all are possible to evaluate in future research.

## CONCLUSIONS

The care Wikipedia invests in the contents of health pages is considerable and the 'live' 'crowd-sourced' and adjudicated peer-reviewing of pages is impressive. The outcomes we were able to use are likely to be only the tip of an 'activity iceberg'. For every person who sought and clicked the reference on the 'intervention' Wikipedia page to seek more information (the primary outcome), many more are likely to have been informed by the page alone. Enriching Wikipedia content is, potentially, a powerful way to improve health literacy and it is possible to test the effects of seeding pages with evidence. This trial should be replicated, expanded and developed.

## Patient and public involvement statement

We did not have patient involvement. However, we did have the involvement of the public. The protocol for this trial[19] was created by a group of Wikipedia users—medical and informatics students. In March 2017, we organised a 1-day meeting to support consultation meeting with students for this trial. This was funded by ESRC (£2.5K of the total described above specifically for this meeting).

The meeting, led by methodologists, also had attendance of representatives of the publisher of the Cochrane Library (John Wiley) and of Wikipedia. However, the primary purpose of the day was to get consultation on how the trial should be undertaken from the perspective of one end-user group of Wikipedia—the students. They have continued to be involved in the drafting and writing of the protocol, the conduct of the trial and this final draft report.

## Trial registration details (registry and number)

This appears at the end of the abstract (including hyperlink). Recognising that registration is important to help consideration by the major journals, we sought this registration early on—at protocol stage. We were informed that we could not register, as we were not randomising human beings. Because Cochrane Schizophrenia's Information Specialist is from Iran, he knew that some local registries do not apply this rule and that key local registries also are uploaded into the international systems—and this includes the registry from Iran—hence why this study is registered there.

**Author affiliations**
[1]Division of Psychiatry and Applied Psychology, University of Nottingham, Nottingham, UK
[2]Nottingham Clinical Trials Unit, University of Nottingham, Nottingham, UK
[3]Health Sciences, Research, John Wiley Ltd, Chichester, UK
[4]Department of Critical Care, East Kent Hospitals University NHS Foundation Trust, Canterbury, UK
[5]General Medicine, Nottingham University Hospitals Healthcare NHS Trust, Nottingham, UK
[6]Orthopaedics, Sherwood Forest Hospitals NHS Foundation Trust, Sutton-in-Ashfield, UK
[7]Emergency Department, Gold Coast University Hospitals, Gold Coast, Queensland, Australia
[8]Faculty Management and Business Science, University of Aalen, Aalen, Germany
[9]The Acute Stroke Unit – Huggett Suite, Royal Lancaster Infirmary, Lancaster, UK
[10]Haematology, Gateshead Health NHS Foundation Trust, Gateshead, UK
[11]Department of Radiology, University Hospitals of Leicester NHS Trust, Leicester, UK
[12]Psychiatry, University of Melbourne, Melbourne, Victoria, Australia
[13]Department of Anaesthesia, Royal Melbourne Hospital, Melbourne, Victoria, Australia
[14]Bristol Medical School, University of Bristol Faculty of Health Sciences, Bristol, UK
[15]Fakultät Gesundheit, Sicherheit und Gesellschaft, Hochschule Furtwangen University, Furtwangen, Germany
[16]Community Recovery Psychiatry, North East London NHS Foundation Trust, London, UK
[17]Liaison Psychiatry, Northumbria Healthcare NHS Foundation Trust, North Shields, UK
[18]Wikimedia UK, Wikimedia Foundation, London, UK
[19]Acute Medicine, Lewisham and Greenwich NHS Trust, London, UK
[20]Accident and Emergency, Epsom and Saint Helier University Hospitals NHS Trust Epsom Hospital, Epsom, UK
[21]Nottingham Ningbo GRADE Centre, Nottingham China Health Institute, The University of Nottingham Ningbo, Ningbo, China
[22]Division of Epidemiology and Public Health, School of Medicine, The University of Nottingham, Nottingham, UK

**Acknowledgements** The research team is grateful to Julie Wood and the Communications and External Affairs team of the Cochrane Collaboration who gave permission for use of the access code for the summaries pages via Google Analytics. Thanks also to Joanne Thomas and Max Goldman (Sense about Science—senseaboutscience.org) who helped with the language within the tables. We also wish to thank James Heilman (Wikipedian, The Wikipedia Open Textbook of Medicine) for his ongoing support in seeding Wikipedia with good evidence.

**Contributors** CEA: envisioned and led the project and gained (modest) funding for it and contributed—in substantial measure—to the planning, conduct and reporting of the trial. AAM: helped led the project and undertook the analyses and contributed—in substantial measure—to the planning, conduct and reporting of the trial.TA, SB, PMB, EC, SC-W, GF, JG, EH, MH, MJ, SN, LP, US, AST, AV, DW, JX: contributed—in substantial measure—to the planning, conduct and

reporting of the trial and greatly assisted data acquisition from John Wiley. JF: contributed—in substantial measure—to the planning, conduct and reporting of the trial and undertook software design (SEED) and particularly helped with data acquisition from Google Analytics. LS: contributed—in substantial measure—to the planning, conduct and reporting of the trial and undertook software design (SEED) and particularly helped with data acquisition from Google Analytics. DT: contributed—in substantial measure—to the planning, conduct and reporting of the trial and provided continual help with the Wikipedia perspective. Contributions: Julie Wood and the Communications and External Affairs team of the Cochrane Collaboration gave permission for use of the access code for the summaries pages via Google Analytics. Joanne Thomas and Max Goldman (Sense about Science—senseaboutscience.org) helped with the language to be used within the tables. James Heilman (Wikipedian, The Wikipedia Open Textbook of Medicine) supported seeding Wikipedia with good evidence.

**Funding** This study was funded by the UK Economic and Social Research Council (ESRC) Impact Accelerator Fund (£7.5K, Grant ID: IAF0034) with intramural support from the University of Nottingham.

**Disclaimer** The researchers have no relationship with the ESRC.

**Competing interests** All posts and edits were undertaken by the user Lena08041993, with any conflicts of interest and affiliation of this account with the Cochrane Schizophrenia Group clearly declared on the user's talk page (https://en.wikipedia.org/wiki/User:Lena08041993). DT is affiliated to the Wikimedia Foundation.

**Patient consent for publication** Not required.

**Ethics approval** This study employed inanimate Cochrane systematic review-generated Wikipedia tables as participants and collected routine data from electronic systems for outcomes. We enquired of the local Ethics Committee and were advised that ethical approval is not required. As such, this study is a prototype for ethical randomised interventions in Wikipedia.

**Provenance and peer review** Not commissioned; externally peer reviewed.

**Data availability statement** All data relevant to the study are available in a public, open access repository. Permanent URL: https://doi.org/10.17605/OSF.IO/K2SP4.

**Open access** This is an open access article distributed in accordance with the Creative Commons Attribution 4.0 Unported (CC BY 4.0) license, which permits others to copy, redistribute, remix, transform and build upon this work for any purpose, provided the original work is properly cited, a link to the licence is given, and indication of whether changes were made. See: https://creativecommons.org/licenses/by/4.0/.

**ORCID iDs**
Clive E Adams http://orcid.org/0000-0003-1628-4020
Paul M Briley http://orcid.org/0000-0002-5372-6505
Mahesh Jayaram http://orcid.org/0000-0002-5352-1075

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
