## [Reviewer comments · BMJ Open]

ARTICLE DETAILS

TITLE (PROVISIONAL)	Adding evidence of the effects of treatments into relevant Wikipedia pages: a randomised trial.
AUTHORS	Adams, Clive; Montgomery, Alan; Aburrow, Tony; Bloomfield, Sophie; Briley, Paul; Carew, Eburn; Chatterjee-Woolman, Suravi; Feddah, Ghaliya; Friedel, Johannes; Gibbard, Josh; Haynes, Euan; Hussein, Mohsin; Jayaram, Mahesh; Naylor, Samuel; Perry, Luke; Schmidt, Lena; Siddique, Umer; Tabaksert, Ayla; Taylor, Douglas; Velani, Aarti; White, Douglas; Xia, Jun

VERSION 1 – REVIEW

REVIEWER	Thomas Shafee La Trobe University, Melbourne, Australia
REVIEW RETURNED	15-Sep-2019

GENERAL COMMENTS	This work describes a randomised experiment on Wikipedia content by adding structured information based on Cochrane reviews to a randomly determined subset of medical articles. It aimed to determine whether addition of this information drove additional traffic to the cited sources. In this aim, it did not detect statistically significant evidence that inclusion of the information increased readership of the sources, however it did gather additional valuable information. The authors appear to have been suitably conservative in their interpretations of the data, whilst extracting useful information. An additional value of this study is as a prototype for ethical randomised interventions in Wikipedia, and as such I would encourage a more thorough description in this section of the methods. My comments for improvement and clarification are listed below using "pagenumber.linenummer" of the pdf provided for review. Methods: ===== p9.48 Experimentation with either wikipedia content, users or readers has been controversial in the past when done wrong (e.g. https://meta.wikimedia.org/wiki/Research_talk:How_role-specific_rewards_influence_Wikipedia_editors%E2%80%99_contribution partly lead to a new WP:NOTLAB guideline, https://en.wikipedia.org/wiki/Wikipedia_talk:What_Wikipedia_is_not/Archive_55#RFC:Wikipedia_Is_Not_a_Laboratory). It would therefore be useful to include a section either in the methods section or Patient and public involvement statement explicitly outlining that the work aimed to comply with specific ethical and policy guidelines e.g. compliance with the terms of use, WP:MEDRS, and WP:NOTLAB guidelines (particularly useful for informing any
---

	future studies wishing to employ similar protocols). In particular:  * None of the content was disruptive to the community or negatively affected articles and so were considered non-controversial edits per WP:NOTLAB * All content added was intended to improve the encyclopedia per WP:TOU * All content was sourced from WP:MEDRS-compliant references * Edits were done by logged-in users (not anonymous ip addresses) p9.59 Since low readability is a repeated limitation noted in table 1, what, if any considerations were made for readability in the tables inserted? p10.37 Consider also including in the outcomes section: retention of the table within the wikipedia page. p10.46 Meaning of "(REF)" unclear p11.21 Now that the study has concluded, are the authors intending to add similar tables to the control wikipedia pages to bring them in line with the intervention group? Results: ===== p11.23 The 12-month intervention timeline is likely sufficient for the requirements of the co-primary measurements in the study. It is possible that other measures (e.g. number of pageviews of the wikipedia pages themselves, or altmetric scores) may have seen larger effects over longer periods of times as editors and readers became more aware of the included tables. The interpretation of the statistical methods has been suitably conservative, with appropriate caveats described for possible confounding factors. p11.44 Consider marking the articles from which added tables were removed in a column of the supplementary summary table. I notice that one table was moved from its original location (life skills) to a different page (Activities of daily living) after editor discussion on the article's talkpage. Was this included as 'removed' or 'retained'. Were any other tables moved? Discussion: ===== p13.46 and p14.3 Regarding the note that "This is the first randomised trial of Wikipedia content": This is true with the provision that there is an unreviewed preprint at https://dx.doi.org/10.2139/ssrn.3039505 "Thompson, N. and Hanley, D., 2018. Science is shaped by wikipedia: Evidence from a randomized control trial". It does do randomised A/B editing of a subset of wikipedia articles but for a very different purpose and analysis. p13.55 Since the altmetric score is a composite measure across multiple output types (news, twitter, blogs, reddit, facebook etc) as well as wikipedia inclusion itself, is it possible to say which of these aspects
--	--

	were primarily responsible for the increased altmetric score? In particular, can the authors confirm that it was not just the contribution of wikipedia inclusion to the altmetric score that causes the observed effect (https://help.altmetric.com/support/solutions/articles/6000060969-how-is-the-altmetric-score-calculated). p14.38 This interpretation seems also in line with wikimedia's own reader motivation and behaviour research https://arxiv.org/abs/1702.05379 p15.46 Consider writing as "500k/month" for clarity p15.50 It may be worth clarifying that more subtle but potentially relevant effects were beyond the scope of the methods used, such as effect on reader behaviour (e.g. https://meta.wikimedia.org/wiki/Research:Which_parts_of_an_article_do_readers_read) or information comprehension. References: ===== p21 References to wikipedia are a mixture of 2015-2018, with only ref 2 linking out to a specific page version. Ideally they should probably point to specific version via "?oldid=00000000" appended to the link. The date version that they point to should probably either be the start of the study (if pointing to a page or policy at the initiation of the study) or pointing to the page version as it exists now (if referring to the current status of a page or policy). Supplementary data: p38 The table in the pdf provided for review has truncated many of the cells (including column headers). Is a csv or equivalent available? If presented as a table split across pages of a pdf, first column should be repeated at start of each page p38.43 ID 42 lists "cannabis", but the wikipedia page is "long term effects of cannabis"
--	---

REVIEWER	John Willinsky Stanford University, USA
REVIEW RETURNED	19-Sep-2019

GENERAL COMMENTS	I have no doubts about the value of learning about how Wikipedia's references and sources are used by readers, nor about the research competencies of the team that carried out the study. I also recognize that clicking on the references in Wikipedia is a good indicator of an interest in further learning. And I am deeply impressed by the educational value of the open sources SEED tool, and can see great merit in assessing whether it leads to click-throughs for Cochrane, suggesting educational interest, and social media sharing, suggesting a fostering of trust in the Wikipedia entry. I did not find the methods set out clearly or the key terms well-defined, and thus struggled to make sense of the design, and thus
--

	suspect some of my points below may be off the mark, but that they at least speak to points in need of clarification. A. "The control group Wikipedia pages did not have a table or reference added – although seven of these pages already had the Cochrane reference employed." Why were there not three groups set up, with (a) the intervention: Cochrane summary table intervention; (b) the "naturally" occurring Cochrane references; and (c) the lack of a table or a reference? Or at least do a further analysis as a check on the results using this sort of division between (b) and (c)? In a similar manner, the Wikipedia editor's deletion of the Cochrane tables from the Wikipedia pages at various points may also have been used as a further checks on conclusions drawn. B. "This intervention resulted in no clear, statistically significant, difference in access to the full review and page views after one year." 1. Without a clear delineation of the design, I found it difficult to see what is being compared with "full review" and "page views" between intervention and control. What is my best guess is that the study compared these results for the 70 Cochrane Schizophrenia reviews without regard to whether the data resulted from Wikipedia click-throughs or not. This may be off. We have no idea what it would take to move the needle, that is, to achieve a significant difference between the Cochrane Schizophrenia reviews in the intervention and the control. Wiley's weblogs would have allowed an identification of the click throughs from Wikipedia, and this could be used to compare for the activity from Wikipedia pages with the summary table vs. the seven with no table but a Cochrane reference (but really that calls for a better design research design). 2. In light of the paywall "protecting" the Cochrane full-text from many of not most readers, and given what I assume about the Wiley weblogs is correct, re: identifying click-throughs from the Wikipedia pages in question to the Cochrane summary and the full-text, why not assess (a) what proportion click on the Cochrane study link in Wikipedia, which leads to the Cochrane summary; (b) what proportion click on "read the full abstract" and (c) what proportion click on "unlock the full review" or "see the full review" and, as a result, either (i) purchase access; (ii) stop there; or, (ii) have subscription access? This subscription information would give us an approximate sense of lay and professional access, as well as stymied access, and highly valued access. C. "With [the] exception of Altmetric score for which there was some evidence of effect" This seems more than a "secondary outcome" in assessing the potential impact of SEED, and more analysis of this activity seems in order, with regard to, for example, whether the social media reference made mention of (a) terms from the intervention table seems in order; (b) link to the Wikipedia page; and (c) to the Cochrane summary page. "All outcomes did favour reviews allocated to the Wikipedia page - there was a consistent 13-36% increase in activity across all
--	---

	findings.” This does not sit well with “there was no evidence of effect on either of the coprimary outcomes... results were similar for all other outcomes, with [the] exception of Altmetric score for which there was some evidence of effect.” D. “Recording of outcome necessitated unusual levels of interest and commitment on the part of the Wikipedia page reader.” As a very minor aspect of this review, the point of this is not clear and certainly the recording of the outcome did not require more work that “recording” any of the other outcomes.
--	---

REVIEWER	Michael Scaffidi Queen's University, Kingston, ON, Canada
REVIEW RETURNED	30-Sep-2019

GENERAL COMMENTS	In this study, the authors aimed to evaluate the effects of enriching Wikipedia content with summary tables from level 1 evidence on the effects of care. To do this, they conducted a two-arm, parallel, open randomized controlled trial of articles on Wikipedia related to schizophrenia. As the authors state in their Discussion, this is indeed a novel approach to the question of addressing the impact of the integration of systematic review on the quality of Wikipedia content. There are, however, several issues that need to be addressed. Comments  1. First and foremost, the research question is not clear: is it an evaluation of the quality of Wikipedia articles? If so, what are the metrics be used? The authors do mention that they look at Altmetrics related to each article but need to explicitly delineate how this is being measured. I recommend that the authors pinpoint exactly what the research question is using the PICO (Population, Intervention, Comparison, Outcome) framework. 2. Stylistically, the article requires a large scale rewrite. Specifically, the authors spend a lot of time expanding on areas that are relatively well known topics (e.g. Wikipedia, Cochrane), which sacrifices space required for more substantial points of discussion. For example, a paragraph description on the editing capacity of Wikipedia in the Background is not required and should be limited to 1 to 2 sentences, at most. More importantly, the Discussion is unfocused and does not highlight what the importance of the study is, nor does it frame the current study in the relevant literature. Please address these issues. Finally, the manuscript features a great deal of informal and unclear language– for example, lines 67-68 describe drugs as “little used”, which should rather be quantitatively defined. 3. From a methodology standpoint, a randomized controlled trial (RCT) is not appropriate, as there are no participants and no clear intervention. Instead, this should be framed as a document review study. Although not impossible to do an RCT in the context of a document study (cf. https://doi.org/10.1136/bmj.a568), it requires a strict inclusion/ exclusion criteria, which were not present in this study. 4. Overall, I think that a serious rewrite of the paper is needed. I believe, however, that the underlying study concept of exploring the insertion of systematic reviews on Wikipedia article quality is an excellent idea that has a great deal of merit.
---

VERSION 1 – AUTHOR RESPONSE

Section	Reviewer ^e	Source – Reviewer #1	Response	Reviewer 1:number
		This work describes a randomised experiment on Wikipedia content by adding structured information based on Cochrane reviews to a randomly determined subset of medical articles. It aimed to determine whether addition of this information drove additional traffic to the cited sources. In this aim, it did not detect statistically significant evidence that inclusion of the information increased readership of the sources, however it did gather additional valuable information. The authors appear to have been suitably conservative in their interpretations of the data, whilst extracting useful information. An additional value of this study is as a prototype for ethical randomised interventions in Wikipedia, and as such I would encourage a more thorough description in this section of the methods.	Thank you for these comments. We added a short sentence to this effect – although did not want to add much text as comments further down this list were likely to lengthen the paper and we did not want to over-labour the point.	R1:1
Methods:	p9 .4 8	Experimentation with either wikipedia content, users or readers has been controversial in the past when done wrong (e.g. https://meta.wikimedia.org/wiki/Research_talk:How_specific_rewards_influence_Wikipedia_editors%E2%80%99_contribution partly lead to a new WP:NOTLAB guideline, https://en.wikipedia.org/wiki/Wikipedia_talk:What_Wikipedia_is_not/Archive_55#RFC:Wikipedia_Is_Not_a_Laboratory).	Thank you for this helpful comment. Before posting the tables we were involved with representatives of the WikiProject Medicine and have now added a statement that outlines our efforts to comply with the Wikipedia’s policies.	R1:2
		It would therefore be useful to include a section either in the methods section or Patient and public involvement statement explicitly outlining that the work aimed to comply with specific ethical and policy guidelines e.g. compliance with the terms of use, WP:MEDRS, and WP:NOTLAB guidelines (particularly useful for informing any future studies wishing to employ similar protocols). In particular:		R1:3 Reference 38 & 39
		* None of the content was disruptive to the community or negatively affected articles and so were considered non-controversial edits per WP:NOTLAB	To emphasise our compliance with Wikipedia policy we have had to add two new reference to Wikipedia policies.	
		* All content added was intended to improve the encyclopedia per WP:TOU		
		* All content was sourced form WP:MEDRS-compliant references		
		* Edits were done by logged-in users (not anonymous ip addresses)		

p9 .5 9	Since low readability is a repeated limitation noted in table 1, what, if any considerations were made for readability in the tables inserted?	We absolutely accept this point – and with some other points further down the list – we have the dilemma of getting the balance between expansive description of this within this document and simple referral to the protocol for this trial and the SEED paper which we have already referenced. However, clearly, from peer comments, this issue did not come across well and we have tried to make things much better. In the ‘Sense about Science’ consultation, the important changes that came up included a change of wording from ‘no statistical significance’ to ‘there was no clear effect’, the addition of a summary on top of the table, and the choice of the 4 column format for the table with outcome names spanning all rows. The details on how we worked to increase readability are described in the protocol, as well as our publication of the SEED tool	R1:4
---------------	--	---	------

		(references 31 & 34). Added: In the design process of our tables we communicated with members of 'Sense about Science'(REFERENCE) and consulted publications of the Cochrane Effective Practice and Organisation of Care group (REFERENCE) in order to increase clarity and readability of the evidence in our tables. More details on how we worked to increase readability are described in the protocol(REFERENCE), as well as our publication of the SEED tool(REFERENCE).	
p1 0. 37	Consider also including in the outcomes section: retention of the table within the wikipedia page.	Thank you for this comment. We did not specify this outcome in our protocol, but as it emerged during the course of the trial we did discuss – as a post hoc finding in the original version but have now made this clearer.	R1:5
p1 0. 46	Meaning of "(REF)" unclear	Sorry – this was a typo	Deleted
p1 1. 21	Now that the study has concluded, are the authors intending to add similar tables to the control wikipedia pages to bring them in line with the intervention group?	This is a good point, and, in truth, we had not planned on doing this. It is a	

		person-power issue. Should the findings have been dramatically favouring the intervention group then there would be an imperative to act but in these constrained times the modest effect – and the effort needed to ‘intervene’ is not clearly cost-effective. We think this trial shows a modest benefit for the seeding of the pages in this way. We think it opens this area to further experimentation for evidence-based seeding into Wikipedia. We would love to keep this going as routine but the good will that generated the randomised trial reported here is a precious commodity. We think this evidence is enough to encourage specifically funded entities into action – but that is not, at present, us.	
Results	p1 1. 23 The 12-month intervention timeline is likely sufficient for the requirements of the co-primary measurements in the study. It is possible that other measures (e.g. number of pageviews of the wikipedia pages themselves, or altmetric scores) may have seen larger effects over longer periods of times as editors and readers became more aware of the included tables. The interpretation of the statistical methods has been suitably conservative, with appropriate caveats described for possible confounding factors.	Thank you for this supportive comment. Page views vary greatly with respect to the time of the year, or when a topic is mentioned in the press. We cannot say if a big increase in pageviews would	

		be equally linked with an increase of people following our links, because people who visit the page due to press articles might have different motives than concerned patients or doctors. Wikipedia pages are 'living'. They are constantly updated and can change rapidly even over the course of a year. Our one-year snapshot was affected slightly by bots inserting pubmed-links or editors re-structuring pages. A longer period might even have led to further approximation of the two groups.	
p1 1. 44	Consider marking the articles from which added tables were removed in a column of the supplementary summary table.	We marked them up, thank you for this comment. We could not get the file to behave in the PDF format produced for final review so have deleted this file and changed the dataset to be one that is available on line as CSV format.	Amended link to supplementary data file R1:6 Permanent URL: https://doi.org/10.17605/OSF.IO/K2SP4
	I notice that one table was moved from its original location (life skills) to a different page (Activities of daily living) after editor discussion on the article's talkpage. Was this included as 'removed' or 'retained'. Were any other tables moved?	Thank you for bringing this up. When this incident happened, the table was not visible on any article for a matter of hours. In light of the 12-month duration of	We have not added text further clarifying this – as we do think any impact on the findings would be negligible - but would be happy to do so if requested

		the trial period, we did not feel this short downtime would have had much impact and the traffic caused by editors of Wikipedia following our link during the discussion of where the table should be placed would also have been negligible. The whole table, including its reference with the links to the Cochrane pages was then posted on the new landing page, and therefore 'retained' for the remainder of the trial. This did not happen any other table.		
Discussion:	p1 3. 46 and p1 4. 3	Regarding the note that "This is the first randomised trial of Wikipedia content": This is true with the provision that there is an unreviewed preprint at https://dx.doi.org/10.2139/ssrn.3039505 "Thompson, N. and Hanley, D., 2018. Science is shaped by wikipedia: Evidence from a randomized control trial". It does do randomised A/B editing of a subset of wikipedia articles but for a very different purpose and analysis.	Oh – thank you so much for finding this. This study had alluded us – despite us having a good look for anything relevant. We have amended the text in our paper to take this into account.	R1:7 – added the word 'placement' three times within the report. and added this reference (# 46)
	p1 3. 55	Since the altmetric score is a composite measure across multiple output types (news, twitter, blogs, reddit, facebook etc) as well as wikipedia inclusion itself, is it possible to say which of these aspects were primarily responsible for the increased altmetric score? In particular, can the authors confirm that it was not just the contribution of wikipedia inclusion to the altmetric score that causes the observed effect (https://help.altmetric.com/support/solutions/articles/6000060969-how-is-the-altmetric-score-calculated).	This is a very good point that we had not fully considered – thank you for raising it. It is perfectly feasible that the elevation in any of a composite scores – or the individual parts of the	

		composite score – was influenced by chance occurrences – and not the placement of evidence within the Wikipedia page. • For example, it is possible that the chance interest of a journalist and a subsequent newspaper article would then boost traffic on a Wikipedia page. However, such chance occurrences should be evenly balanced between 'treated' pages and the control group.• It is feasible, in this small trial, that such chance increases in Wikipedia activity are imbalanced into	R1:8
--	--	--	-------------

		either the treatment or control group.  • It is also possible that, if balanced, which is more likely the case, they serve to negate the differences that occur through randomisation. It is also possible that the elevation of the Altmetric is simply caused by elevation of the Wikipedia scoring within the altmetric ratings. Because of this reviewer's comments we revisited the Altmetric scores. Inspection of the contributing parts of the composite Altmetric score (please see Supplementary file - Permanent URL: https://doi.org/10.17605/OSF.IO/K2SP4) give no indication that the Wikipedia sub-score is simply causing the elevation in Altmetric ratings. The elevation seems more linked to micro-blogging such as Twitter activity. We added a sentence in the	
--	--	---	--

		Discussion to this effect.	
	p1 4. 38	This interpretation seems also in line with wikimedia's own reader motivation and behaviour research https://arxiv.org/abs/1702.05379	Thank you.
	p1 5. 46	Consider writing as "500k/month" for clarity	Done R1:9
	p1 5. 50	It may be worth clarifying that more subtle but potentially relevant effects were beyond the scope of the methods used, such as effect on reader behaviour (e.g. https://meta.wikimedia.org/wiki/Research:Which_parts_of_an_article_do_readers_read) or information comprehension.	Thank you – this is really helpful and we have clarified this by adding a sentence to the outcomes section R1:10
References:	p2 1	References to wikipedia are a mixture of 2015-2018, with only ref 2 linking out to a specific page version. Ideally they should probably point to specific version via "?oldid=00000000" appended to the link. The date version that they point to should probably either be the start of the study (if pointing to a page or policy at the initiation of the study) or pointing to the page version as it exists now (if referring to the current status of a page or policy).	Thank you for this comment, we will provide these permanent links in the revised version of this article. References 1, 7, and 28 have been updated.
Supplementary data:	p3 8	The table in the pdf provided for review has truncated many of the cells (including column headers). Is a csv or equivalent available?	Of course we can make this available – the PDF version was very unserviceable – as this reviewer recognised. We are not sure if the XLS file would, in the published version, remain intact – and not be PDF-ed but, in any case, we have created a CSV file and put this online. Link to data created
		If presented as a table split across pages of a pdf, first column should be repeated at start of each page	I am sorry – we will ensure this does not happen CSV file now available. R1:6
	p3 8. 43	ID 42 lists "cannabis", but the wikipedia page is "long term effects of cannabis"	We thought this to be the best target page.

Source – reviewer #2	Response	Reviewer2:number
I have no doubts about the value of learning about how Wikipedia's references and sources are used by readers, nor about the research competencies of the team that carried out the study. I also recognize that clicking on the references in Wikipedia is a good indicator of an interest in further learning. And I am deeply impressed by the educational value of the open sources SEED tool, and can see great merit in assessing whether it leads to click-throughs for Cochrane, suggesting educational interest, and social media sharing, suggesting a fostering of trust in the Wikipedia entry.	Thank you	
I did not find the methods set out clearly or the key terms well-defined, and thus struggled to make sense of the design, and thus suspect some of my points below may be off the mark, but that they at least speak to points in need of clarification.	We are sorry this reviewer and, therefore, readers would have struggled to make sense of the design. We have revisited the methods section because of this and other comments above and below and hope they are now clearer.	We have edited throughout the methods sections (R2:1 onwards) – but added subtitles that are clearer e.g. R2:2 and the PICO box at R3:2
A. "The control group Wikipedia pages did not have a table or reference added – although seven of these pages already had the Cochrane reference employed." Why were there not three groups set up, with (a) the intervention: Cochrane summary table intervention; (b) the "naturally" occurring Cochrane references; and (c) the lack of a table or a reference? Or at least do a further analysis as a check on the results using this sort of division between (b) and (c)? In a similar manner, the Wikipedia editor's deletion of the Cochrane tables from the Wikipedia pages at various points may also have been used as a further checks on	We intended from the beginning to carry out a stratified randomisation over all eligible reviews. Including the subgroup of pages that already had a reference in as a separate group would not have added to the randomised trial – as that group is not randomised. We thought undertaking randomisation across even 'partially treated' eligible pages would fit with pragmatic trial design (1). In light of the WP:NOTLAB(2) policy we aimed to be non-disruptive, and hence made no changes to existing references in control articles. Our 'intervention' citation included links to not only full-text but also	Tried to clarify this issue of the pragmatic design: R2:3 + reference #43 Provided some support for non-intervention into current pages: R2:4; R2:5

conclusions drawn.	the universally accessible summary whereas the control serendipitously referenced pages only contained a reference to the full text.	
B. “This intervention resulted in no clear, statistically significant, difference in access to the full review and page views after one year.” 1. Without a clear delineation of the design, I found it difficult to see what is being compared with “full review” and “page views” between intervention and control. What is my best guess is that the study compared these results for the 70 Cochrane Schizophrenia reviews without regard to whether the data resulted from Wikipedia click-throughs or not. This may be off. We have no idea what it would take to move the needle, that is, to achieve a significant difference between the Cochrane Schizophrenia reviews in the intervention and the control. Wiley’s weblogs would have allowed an identification of the click throughs from Wikipedia, and this could be used to compare for the activity from Wikipedia pages with the summary table vs. the seven with no table but a Cochrane reference (but really that calls for a better design research design).	We really are sorry that we have caused the confusion. If this reviewer has had problems - clearly others will too and we have made efforts to try to offset this throughout the paper. We hope clarification in the methods has helped and we have added a PICO table as part of this.	We have edited throughout the methods sections (R2:1 onwards) – but added subtitles that are clearer e.g. R2:2 and the PICO box at R3:2
2. In light of the paywall “protecting” the Cochrane full-text from many of not most readers, and given what I assume about the Wiley weblogs is correct, re:	This is an important point – and a sore point to this particular writer. The paywall issue is real and that is why we used the Google Analytics data (concerning use of the free full	

identifying click-throughs from the Wikipedia pages in question to the Cochrane summary and the full-text, why not assess (a) what proportion click on the Cochrane study link in Wikipedia, which leads to the Cochrane summary; (b) what proportion click on “read the full abstract” and (c) what proportion click on “unlock the full review” or “see the full review” and, as a result, either (i) purchase access; (ii) stop there; or, (ii) have subscription access? This subscription information would give us an approximate sense of lay and professional access, as well as stymied access, and highly valued access.	text abstract) as well as the data relating to the not-so-free full text PDF as produced by Wiley. We agree some figure on ‘stymied’ access would have been really good but we just do not have those data and are not sure that anyone has. We have re-worded a passage and added a short sentence regarding halted access.	R2:6
C. “With [the] exception of Altmetric score for which there was some evidence of effect” This seems more than a “secondary outcome” in assessing the potential impact of SEED, and more analysis of this activity seems in order, with regard to, for example, whether the social media reference made mention of (a) terms from the intervention table seems in order; (b) link to the Wikipedia page; and (c) to the Cochrane summary page.	Thank you for suggesting this. We did not outline this in-depth analysis of altmetrics in our protocol – and felt it would be too much post-hoc analyses should we then consider finer details after collecting the data. The breakdown of the altmetrics is in the data file and this affords opportunity to hypothesis-generation in future work. In theory we could have tracked the altmetric components one by one and analysed what they were mentioning but this was beyond our resource – but is a good point for a new study. This type of work is perfect for the detail exploration that would be expected of a relevant PhD – but working with the good will of all concerned – such interesting detail could not be explored in this study. Have added a short text about component parts of Altmetric score.	R3:1
“All outcomes did favour reviews allocated to the Wikipedia page - there was a consistent 13-36% increase in activity across all findings.” This does not sit well with “there was no evidence of effect on either of the coprimary outcomes... results were similar for all other outcomes, with [the] exception of Altmetric score for which there was some evidence	We were acutely conscious of this – having spent much of our lives trying not to be biased and over-stressing modest findings. We were trying to get a balance in stating that most findings were indeed not reaching the conventional levels of statistical significance but all indicators moved and all towards some positive effect of use of Wikipedia tables. As one of your peer reviewers point	R2:7

of effect.”	out we have tried to tread a cautious line – but have slightly modified the language again to further retreat from the danger of over-emphasising hypothesis over fact.	
D. “Recording of outcome necessitated unusual levels of interest and commitment on the part of the Wikipedia page reader.”	Thank you for pointing this out. We have deleted the word ‘recording’ altogether.	
As a very minor aspect of this review, the point of this is not clear and certainly the recording of the outcome did not require more work than “recording” any of the other outcomes.		

Source – Reviewer #3	Response	Reviewer3:number
As the authors state in their Discussion, this is indeed a novel approach to the question of addressing the impact of the integration of systematic review on the quality of Wikipedia content.	Thank you.	
First and foremost, the research question is not clear: is it an evaluation of the quality of Wikipedia articles? If so, what are the metrics be used? The authors do mention that they look at Altmetrics related to each article but need to explicitly delineate how this is being measured. I recommend that the authors pinpoint exactly what the research question is using the PICO (Population, Intervention, Comparison, Outcome) framework.	We are sorry that we have not been clear. This is also a comment from peer reviewer #2 and we have taken this most seriously and undertaken careful re-drafting to increase clarity – with particular attention to Altmetrics. Throughout the article we have tried to really clarify the text and methods and have included a new PICO table. Altmetrics is calculated different ways – depending on the company. We have referenced John Wiley’s description of how they use them. The specific formula is not described in our piece as we felt this would be too detailed but tried to	PICO box at R3:2 R3:1 and Reference #40

	clarify how this score is commonly used – albeit tailored by each company – as a measure of reach. Peer reviewer #2 requested clarification on whether the Altmetric was just a proxy for activity generated by Wikipedia – and we hope we have clarified this as not being the case. From the data in the supplementary file, and from the weighting within the calculation of the altmetric – the micro-blogging media carries most weight – and, in this case, most activity	
2. Stylistically, the article requires a large scale rewrite. Specifically, the authors spend a lot of time expanding on areas that are relatively well known topics (e.g. Wikipedia, Cochrane), which sacrifices space required for more substantial points of discussion. For example, a paragraph description on the editing capacity of Wikipedia in the Background is not required and should be limited to 1 to 2 sentences, at most. More importantly, the Discussion is unfocused and does not highlight what the importance of the study is, nor does it frame the current study in the relevant literature. Please address these issues. Finally, the manuscript features a great deal of informal and unclear language– for example, lines 67-68 describe drugs as “little used”, which should rather be quantitatively defined.	Oh – we are sorry that the style was a problem and do not want this to impede dissemination. We have redrafted the background and discussion. We have cut out most of the text on Cochrane from the background – although are a bit worried about leaving this too thin for those who do not know the organisation. We did not reduce much the information on Wikipedia and how it works as we thought this may be less familiar to readers – it was to us. We have tried to focus the discussion much more and tried to set this study alongside relevant literature Framing the study in the relevant literature: Table 1 summarises related	We have not marked this up as there is too much to mark. We did endeavour to clarify throughout – probably especially by shortening the background – deleting some about Cochrane. This is also difficult to mark up but we have tried to do this in the Discussion paragraph starting “There is little similar literature to contextualise this work” R3:3 Have added text “(trifluoperazine – a less used antipsychotic e.g 3529 ± 198 prescriptions/month - figures are for 2018, NHS England(29); chlorpromazine – a old widely-used antipsychotic drug 22386 ±803 prescriptions/month; paliperidone – a expensive new antipsychotic drug 853 ±

	literature with respect to medical content of Wikipedia. We hope we have improved the discussion of the current study in respect of the few trials existing in this area. We have removed the 'little used' and added numbers into this section – although the figures we have are limited by region of prescription and date. There is a danger of making this statement wordy but it is certainly improved for adding more verifiable data.	34 prescriptions/month, and one important talking therapy - cognitive behavioural therapy)." R:3:4
3. From a methodology standpoint, a randomized controlled trial (RCT) is not appropriate, as there are no participants and no clear intervention. Instead, this should be framed as a document review study. Although not impossible to do an RCT in the context of a document study (cf. https://doi.org/10.1136/bmj.a568), it requires a strict inclusion/ exclusion criteria, which were not present in this study.	This is a point we have to disagree on. It is entirely feasible to undertake randomisation of participants that are not people. We did have participants and are sorry we have not clarified this well enough and have tried to amend this as encouraged to do by reviewers #2 and #3 above. The participants were relevant Wikipedia pages. We did have strict inclusion criteria and undertook considerable preparation to ensure this was so – as outlined in the paper. This allocation of non-sentient participants in in keeping with the early tradition of randomisation emerging from agricultural experiments – in which	PICO box at R3:2

	it is still heavily employed ¹ or from employment practice ² , inanimate objects in health ³ , information ⁴ or the form of information ⁵ ⁶	
4. Overall, I think that a serious rewrite of the paper is needed. I believe, however, that the underlying study concept of exploring the insertion of systematic reviews on Wikipedia article quality is an excellent idea that has a great deal of merit.	We hope that you find the substantial re-drafting we have undertaken improves the paper – we think it does and thank you for your direction.	

1. Thorpe KE, Zwarenstein M, Oxman AD, Treweek S, Furberg CD, Altman DG, et al. A pragmatic-explanatory continuum indicator summary (PRECIS): a tool to help trial designers. *J Clin Epidemiol.* 2009 May;62(5):464–75.

2. Wikipedia:What Wikipedia is not. In: Wikipedia [Internet]. 2019 [cited 2019 Nov 14]. Available from: https://en.wikipedia.org/w/index.php?title=Wikipedia:What_Wikipedia_is_not&oldid=926025458

1

https://www.povertyactionlab.org/sites/default/files/documents/Agriculture%20deJanvry%20Sadoulet%20Suri%20July%202016_1.pdf

² https://scholar.harvard.edu/files/lee/files/scott_lee_do-gooders_latest.pdf

³ <https://www.bmj.com/content/341/bmj.c6801>

⁴ <https://bmcmmedicine.biomedcentral.com/articles/10.1186/s12916-019-1330-9>

⁵ <https://bmjopen.bmj.com/content/6/3/e010509>

⁶ <https://bmjopen.bmj.com/content/9/4/e025380>

VERSION 2 – REVIEW

REVIEWER	Thomas Shafee La Trobe University, Australia
REVIEW RETURNED	01-Dec-2019

GENERAL COMMENTS	The authors have addressed all of my review comments. In particular, the methods section (and associated supplementary materials) have been very much improved, as has readability. I think that these have improved its usefulness to readers looking to understand the overall conclusions of the paper as well as any looking to replicate or extend on it in future. Very minor: The inclusion of "effect" twice in the title scans oddly, so an alternative wording might be better (however if there are specific conventions in the language of such study titles, then it is not a great problem)
---

REVIEWER	John Willinsky Stanford University
REVIEW RETURNED	04-Dec-2019

GENERAL COMMENTS	I'm impressive and appreciative of the extent to which the authors addressed the concerns raised by the three reviewers, as it turns the paper into a much more helpful contribution to the field. With all that was improved in the paper, I was then struck this time (with apologies for not noting it before) by the "conclusion" (in the abstract and at the end of the paper) which seems to be as much as a conclusion about the experience of working with Wikipedia (being impressed by its balance, care, reach, etc.), rather than a conclusion to the study, its findings and its implications. The paper's final sentence does return to the study itself but is a bit hard to follow: "The outcomes we were able to use are likely to be only the tip of the 'activity iceberg' and placing evidence within Wikipedia seems likely to raise the profile of – in this case – the effects of care."
--

VERSION 2 – AUTHOR RESPONSE

Reviewer #1 - to my embarrassment - was absolutely correct. The title we had been using was truly rubbish - a real not-seeing-the-forest-for-the-trees situation - so I have amended this and hope the new title is better. [Marked up copy - yellowed text]

Reviewer #2 - this is also a good point. In the conclusion, we have had statements to the experience of working with Wikipedia and this is not directly from the numeric findings of the trial. However, we did want to keep something to that effect in. Many disparaging things are said of Wikipedia by academic medicine but that our experience of the system was of an organisation that took fairness very seriously and, although the peer review system is very different to that of standard journals, it did seem to work, it was transparent and carefully managed by highly professional editors.

What we have done is slightly diluted the emphasis and reordered and - hopefully - clarified the wording. The in-praise-of-Wikipedia-systems sentence is moved to the start of the conclusion. We had to keep it as it was, indeed, a conclusion we made. The statements relating to the numeric findings are all moved together after this.